# A Review of the Genus *Homidia* (Collembola, Entomobryidae) in China Informed by COI DNA Barcoding, with the Description of Three New Species [note 1]

**DOI:** 10.3390/insects16090974

**Published:** 2025-09-17

**Authors:** Xiaowei Qian, Yu Fu, Yitong Ma

**Affiliations:** School of Life Sciences, Nantong University, Nantong 226000, China; qianxw@ntu.edu.cn (X.Q.); 2309110043@stmail.ntu.edu.cn (Y.F.)

**Keywords:** chaetotaxy, Chongqing, molecular marker, springtails, new species

## Abstract

**Simple Summary:**

The family Entomobryidae Tömösvary, 1882, is the largest family in Collembola, with around 2500 species in the world. It is characterised by the reduced prothorax, long antennae and furcula, and 4th abdominal segment much longer than 3rd. The genus *Homidia*, belonging to Entomobryidae, is mainly distributed in China. To date, 60 species of *Homidia* have been reported from China and account for approximately 71% of all known species of the genus. Here, the sequences of COI for ten *Homidia* species are provided, a neighbour-joining tree of *Homidia* is presented, three new species are described from Chongqing, China, and the taxonomic statuses of some species are discussed.

**Abstract:**

The genus *Homidia* contains 84 species of which 60 have been reported from China. The sequence of COI for ten *Homidia* species are provided and a neighbour-joining tree is presented. Three new species of *Homidia* are described from Chongqing Municipality, China. *Homidia wuxiensis* sp. nov. is characterised by its colour pattern and chaetotaxy of Abd. IV; *Homidia pseudochroma* sp. nov. by some expanded post-labial chaetae and chaetotaxy of dorsal head and Abd. II–IV and *Homidia yangi* sp. nov. by its colour pattern. Based on similarities in COI sequences and morphology, we designate *Homidia linhaiensis* (Shi, Pan & Qi), as a junior synonym of *Homidia tiantaiensis* (Chen & Li).

## 1. Introduction

The genus *Homidia* was established as a subgenus of *Entomobrya* by Börner based on the presence of the inner spines at the base of the dens [1]. Denis considered this characteristic significant enough to raise *Homidia* to a generic level [2]. The genus is also characterised by the presence of a row of macrochaetae in eyebrow-like formation on the anterior part of 4th abdominal segment, a bidentate mucro with the subapical tooth much larger than the apical and the absence of scales [3].

To date, 84 species of the genus *Homidia* have been described worldwide and 60 species of them have been reported from China (Table 1) [4].

Sixteen species of *Homidia* have been reported from the Korean Peninsula, fourteen from Japan, seven from Vietnam, six from the United States of America, two from India and Thailand, one from Bangladesh, Indonesia, Malaya and Singapore (Table 2).

The colour pattern is an important element in the taxonomy of the genus and it is intraspecifically stable but it varies between different species in general. In fact, some species were named based on the colour pattern. The chaetotaxy of dorsal body is another important character, but it is interspecifically conservative. Other elements, such as the labial and post-labial chaetotaxy, tenent hair and ventral tube, are also useful in the taxonomy of *Homidia*. With the development of molecular methods, some molecular markers, such as COI, are used in its taxonomy. The COI sequences for 25 *Homidia* species are available in GenBank. Here, we provide the COI sequences for ten additional species of *Homidia* and present an expanded neighbour-joining tree of available COI sequences. Some species are discussed and three new species are described from Chongqing (China).

## 2. Material and Methods

### 2.1. Taxon Sampling and Specimens Examinations

Specimens were collected with an aspirator from leaf litter and stored in 99% alcohol. They were mounted on glass slides in Marc André II solution and were studied with a Leica DM2500 phase contrast microscope (Wetzlar, Hessen, Germany). Photographs were taken using a Leica DFC300 FX digital camera (Wetzlar, Hessen, Germany) mounted on the microscope and edited with Photoshop CS2 9.0 version (Adobe Inc., San Jose, CA, USA).

The nomenclature of the dorsal macrochaetotaxy of the head and interocular chaetae follows Jordana and Baquero (2005) and Mari-Mutt (1979, 1986) [5,6,7]. Labial chaetae are designated following Gisin (1964) [8]. Labral chaetae follows Szeptycki (1973) and trunk dorsal chaetotaxy follows Szeptycki (1979) [9,10].

The abbreviations used are as follows. Ant.—antennal segment(s); Th.—thoracic segment(s); Abd.—abdominal segment(s); mac—macrochaeta(e); mes—mesochaeta(e); ms—specialised microchaeta(e); sens—specialised ordinary chaeta(e).

### 2.2. DNA Extraction and Amplification

DNA was extracted by using an Ezup Column Animal Genomic DNA Purification Kit (Sangon Biotech, Shanghai, China) following the manufacturer’s standard protocols. Amplification of a 658 bp fragment of the mitochondrial COI gene was carried out using a Prime Thermal Cycler (TECHNE, Bibby Scientific Limited, Stone, Staffordshire, UK), performed in 25 μL volumes using Premix Taq polymerase system (Takara Bio, Otsu, Shiga, Japan). The primers and polymerase chain reaction (PCR) programs followed Greenslade et al. (2011) [11]. All PCR products were checked using a 1% agarose gel electrophoresis. Successful products were purified and sequenced on an ABI 3730XL DNA Analyser (Applied Biosystem, Foster City, CA, USA). All experiments were completed by Shenggong (Shanghai, China). The GenBank accession numbers are provided in Table 3.

### 2.3. Neighbour-Joining Tree

DNA sequences were assembled using Sequencher 4.5 (Gene Codes Corp., Ann Arbor, MI, USA) and then deposited in GenBank. Sequences were aligned using ClustalW implemented in MEGA 5.1 (Tamura et al., 2011) [12] with default settings. Pairwise genetic distances were analysed in MEGA 5.1 under the Kimura 2-parameter (K2-P) model (Kimura, 1980) [13]. The neighbour-joining tree was presented in the following discussion.

## 3. Results

### 3.1. Taxonomy

Class Collembola Lubbock, 1873 [14].

Order Entomobryomorpha Börner, 1913 [15].

Family Entomobryidae Schäffer [16].

Genus *Homidia* Börner, 1906 [1].

Type species: *Homidia cingula* (Börner, 1906: 174) [1].

#### 3.1.1. *Homidia wuxiensis* sp. nov.

Figure 1, Figure 2, Figure 3, Figure 4, Figure 5 and Figure 6, Table 4.

Type material

Holotype: Female on slide, China, Chongqing Municipality, Wuxi County, the Yintiaoling National Nature Reserve, the Hongqi Protection Station, 31°30′33″ N, 109°49′10″ E, 1129.10 m a.s.l., 21 July 2024, Yitong Ma leg. Paratypes: one female on slide, same collection data as holotype; three females on slides, China, Wuxi County, the Yintiaoling National Nature Reserve, the Lanying Protection Station, Xi’an Village, 31°24′01″ N, 109°51′51″ E, 1625.63 m a.s.l., 26 July 2024, Yitong Ma leg.

Etymology

Named after its locality: Wuxi County.

Diagnosis

Th. II–III brown; brown pigment present at basal parts of Ant. III–IV; labial base with MReL_1_L_2_, e smooth, other ciliate; Ant. IV with 19–25 anterior and 13–20 posterior mac; dens with 37–63 smooth inner spines.

Description

Measurement: Body length up to 3.28 mm.

Colour: Ground colour pale yellow; eye patches dark blue; Th. II–III brown; brown pigment present at basal parts of Ant. III–IV, medial and posterior parts of Abd. IV; anterior part of head, coxae of middle and hind legs, tibiotarsi of fore and middle legs with a little scattered brown pigment (Figure 1).

Head: Antenna not annulated and 1.05–1.10 times length of body. Ratio of Ant. I–IV as 1.00/1.25–1.29/1.00–1.10/2.00. Distal part of Ant. IV with many sensory chaetae and normal ciliate chaetae, apical bulb bilobed (Figure 2A). Ant. III sense organ with two rods, two spiny guard sensilla, smooth blunt sens and ciliated chaetae (Figure 2B). Ant. II with 3–4 rods apically (Figure 2C). Prelabral and labral chaetae as 4/5, 5, 4, all smooth, a2 and b2 slightly shorter than middle ones, labral papillae not clearly seen (Figure 2D). Eyes 8 + 8, G and H smaller than others, interocular chaetae as p, r, t mes. Dorsal chaetotaxy of head with 5–6 antennal (An), five median (M) and eight sutural (S) mac (Figure 2E). Basal chaeta on maxillary outer lobe slightly thicker than apical one; sublobal plate with three smooth chaetae-like processes (Figure 2F). Lateral process (l. p.) of labial palp E differentiated with tip not reaching apex of papilla E (Figure 2G). Labial base with MReL_1_L_2_, e smooth, other ciliate, R 0.60–0.79 length of M (Figure 2H).

Thorax: Tergal ms formula on Th. II–Abd. V as 1, 0/1, 0, 1, 0, 0, sens as 2, 2/1, 2, 2, 2, 3 (Figure 3A, Figure 4 and Figure 5A,C). Th. II with 5–8 medio-medial (m1, m2, m2i, m2i2, 1–4 unnamed mac), three medio-sublateral (m4, m4i, m4p), 46–50 posterior mac. Th. III with 43–49 mac (Figure 3A). Coxal macrochaetal formula as 3/4 + 1, 3/4 + 2 (Figure 3B–D). Trochanteral organ with 86–111 smooth chaetae (Figure 3E). Tenent hair clavate, 0.86–1.00 length of inner edge of unguis; unguis with 3–4 inner teeth, basal pair located at 0.42–0.44 distance from base of inner edge of unguis, distal unpaired teeth at 0.67–0.68 and 0.85–0.86 distance from base, respectively, most distal one very faint and usually absent; unguiculus lanceolate, outer edge slightly serrate (Figure 3F,G).

Abdomen: Range of Abd. IV length as 7.67–7.86 times as dorsal axial length of Abd. III. Abd. I with 11 (a1a, a1–3, m2i, m2–4, m4i, m4p and a5 mac). Abd. II with six (a2, a3, m3, m3e, m3ea, m3ep) central, one (m5) lateral mac. Abd. III with two (a2, m3) central, four (am6, pm6, m7a, p6) lateral mac (Figure 4). Abd. IV with two normal sens, 19–25 anterior, 13–25 posterior and 19–27 lateral mac or mes (Figure 5A,B). Abd. V with three sens (Figure 5C). Anterior face of ventral tube with 58–77 ciliate chaetae, 3+3 of them as mac, line connecting proximal (Pr) and external-distal (Ed) mac oblique to median furrow (Figure 6A); posterior face with 4–5 smooth chaetae apically (Figure 6B); lateral flap with 6–10 smooth and 26–33 ciliate chaetae (Figure 6C). Manubrial plate dorsally with 13–16 ciliate mac and three pseudopores (Figure 6D); ventrally with 39–48 ciliate chaetae on each side (Figure 6E). Dens with 37–63 smooth inner spines (Figure 6F). Mucro bidentate with subapical tooth larger than apical one; tip of basal spine reaching apex of subapical tooth; distal smooth section of dens almost equal to mucro in length (Figure 6G).

Remarks

The new species is characterised by its colour pattern and chaetotaxy of Abd. IV and is mostly similar to the species *H. anhuiensis* Li & Chen, 1997 [17] and *H. speciosa* Szeptycki, 1973 [9], but there are some differences between them, such as the colour pattern on head, labial chaetotaxy, number of central mac on Abd. IV posteriorly and other characters. The detailed characteristic comparisons are listed in Table 4.

**Table 4 insects-16-00974-t004:** Main differences among the new species and similar species of *Homidia*.

Characters	*H. wuxiensis* sp. nov.	*H. anhuiensis*	*H. speciosa*
Colour pattern on head	yellow mainly	black entirely	black entirely
Labial chaetotaxy	MReL_1_L_2_	MREL_1_L_2_	MReL_1_L_2_
Central mac on Abd. IV anteriorly	19–25	9	10–11 *
Central mac or mes on Abd. IV posteriorly	13–20	6–7	5–6
Dental spines	37–63	12–21	28–43

* based on Zhou & Ma, 2023 [18].

#### 3.1.2. *Homidia pseudochroma* sp. nov.

Figure 7, Figure 8, Figure 9, Figure 10, Figure 11 and Figure 12, Table 5.

Type material

Holotype: Female on slide, China, Chongqing Municipality, Wuxi County, the Yintiaoling National Nature Reserve, the Hongqi Protection Station, 31°30′33″ N, 109°49′10″ E, 1129.10 m a.s.l., 21 July 2024, Yitong Ma leg. Paratypes: Three females on slides, same collection data as holotype.

Etymology

Named after its similarity in colour pattern to *H. chroma* Pan & Yang.

Diagnosis

Ant. II–IV and Abd. V brown; posterior margin of Abd. III and medial and posterior parts of Abd. IV with narrow transverse chrome to brown pigmented bands; labial base with MM_1_ReL_1_L_2_, M_1_ rarely duplicate, e smooth, other ciliate, G_1–2_, X_3_ and H_2_ slightly expanded, G_3–4_, X and H_3–4_ strongly expanded in post-labial area; dens with 46–58 smooth inner spines.

Description

Measurement: Body length up to 3.11 mm.

Colour: Ground colour pale white; eye patches dark blue; Ant. II–IV and Abd. V brown; posterior margin of Abd. III and medial and posterior parts of Abd. IV with narrow transverse chrome to brown pigmented bands; little scattered brown or purple pigment present on tibiotarsi and lateral part of Th. II–Abd. II (Figure 7).

Head: Antenna not annulated and 0.48–0.59 times length of body. Ratio of Ant. I–IV as 1.00/1.35–1.59/1.00–1.33/1.77–2.27. Distal part of Ant. IV with many sensory chaetae and normal ciliate chaetae, apical bulb bilobed (Figure 8A,B). Ant. III sense organ with two rods, two spiny guard sensilla, smooth blunt sens and ciliated chaetae (Figure 8C). Ant. II with 3–4 rods apically (Figure 8D,E). Prelabral and labral chaetae as 4/5, 5, 4, all smooth, a2 and b2 slightly shorter than middle ones, labral papillae not clearly seen (Figure 8F). Eyes 8 + 8, G and H smaller than others, interocular chaetae as p, r, t mes. Dorsal chaetotaxy of head with four mac in antennal (An) area, five mac and one mes in median (M) area and 10 mac and 0–2 mes in sutural (S) area (Figure 8G–I). Basal chaeta on maxillary outer lobe slightly thicker than apical one; sublobal plate with three smooth chaetae-like processes (Figure 9A). Lateral process (l. p.) of labial palp E differentiated with tip almost reaching apex of papilla E (Figure 9B). Labial base with MM_1_ReL_1_L_2_, M_1_ rarely duplicate, e smooth, other ciliate, R 0.56–0.64 length of M; G_1–2_, X_3_ and H_2_ slightly expanded, G_3–4_, X and H_3–4_ strongly expanded in post-labial area (Figure 9C).

Thorax: Tergal ms formula on Th. II–Abd. V as 1, 0/1, 0, 1, 0, 0, sens as 2, 2/1, 2, 2, 2, 3 (Figure 10A, Figure 11 and Figure 12A,C). Th. II with four medio-medial (m1, m2, m2i, m2i2) mac, three medio-sublateral (m4, m4i, m4p), 36–41 posterior mac. Th. III with 54–57 mac (Figure 10A). Coxal macrochaetal formula as 3/4 + 1, 3/4 + 2 (Figure 10B–D). Trochanteral organ with 46–68 smooth chaetae (Figure 10E). Tenent hair clavate, 0.88–1.02 length of inner edge of unguis; unguis with 3–4 inner teeth, basal pair located at 0.39–0.42 distance from base of inner edge of unguis, distal unpaired teeth at 0.65–0.69 and 0.87–0.88 distance from base, respectively, most distal one very faint and usually absent; unguiculus lanceolate, outer edge slightly serrate (Figure 10F,G).

Abdomen: Range of Abd. IV length as 8.26–10.8 times as dorsal axial length of Abd. III. Abd. I with 12 (a1a, a1–3, m2i, m2–5, m4i, m4p, a5) mac. Abd. II with seven (a2, a3, m3, m3e, m3ea, m3ep, me3i) central, one (m5) lateral mac. Abd. III with two (a2, m3) central, five (am6, pm6, m7a, p6, p7) lateral mac (Figure 11). Abd. IV with two normal sens, 21–22 anterior, 14–18 (rarely 23) posterior and 32–37 lateral mac or mes (Figure 12A,B). Abd. V with three sens (Figure 12C). Anterior face of ventral tube with 40–42 ciliate chaetae, 3+3 of them as mac, line connecting proximal (Pr) and external-distal (Ed) mac oblique to median furrow (Figure 13A); posterior face with 5–7 smooth chaetae apically (Figure 13B); lateral flap with about seven smooth and 20 ciliate chaetae (Figure 13C). Manubrial plate dorsally with 12–14 ciliate mac and three pseudopores (Figure 13D); ventrally with 38–42 ciliate chaetae on each side (Figure 13E). Dens with 46–58 smooth inner spines (Figure 13F). Mucro bidentate with subapical tooth larger than apical one; tip of basal spine reaching apex of subapical tooth; distal smooth section of dens almost equal to mucro in length.

Remarks

The new species is characterised by some expanded post-labial chaetae, the dorsal chaetotaxy of the head and Abd. II–IV, such as m3ei mac on Abd. II and p7 mac on Abd. IV. It is similar to *H. pentachaeta* Li & Christiansen, 1997 in the chaetotaxy of Abd. III, but the colour pattern, dorsal head chaetotaxy and other characters can be used to separate them [19]. It is also similar to *H. chroma* Pan & Yang, 2019 and *H. obliquistria* Ma & Pan, 2017 in the colour pattern, but there are many differences between them, such as the chaetotaxy of head and Abd. II–IV, shape of chaetae of post-labial area and other characters [20,21]. The detailed characteristic comparisons are listed in Table 5.

#### 3.1.3. *Homidia yangi* sp. nov.

Figure 14, Figure 15, Figure 16, Figure 17, Figure 18 and Figure 19, Table 6.

Type material

Holotype: Female on slide, China, Chongqing Municipality, Wuxi County, the Yintiaoling National Nature Reserve, the Guanshan Protection Station, 31°32′14″ N, 109°41′53″ E, 2168.92 m a.s.l. 28 July 2024, Yitong Ma leg. Paratypes: two females on slides, same collection data as holotype.

Etymology

Named after Mr. Zhiming Yang, whose help is essential to the research of the biological diversity of the Yintiaoling National Nature Reserve.

Diagnosis

Th. III and Abd. III brown; Abd. IV almost brown entirely; Labial base with MReL_1_L_2_, e smooth, other ciliate, L_1_ rarely smooth; Abd. IV with nine (rarely 12) anterior and six posterior mac; dens with about 36 smooth inner spines.

Description

Measurement: Body length up to 2.35 mm.

Colour: Ground colour pale yellow; eye patches dark blue; Ant. I–IV, Th. III and Abd. III brown; Abd. IV almost brown entirely; head, legs, Th. II, Abd. I–II and V with brown pigment; scattered brown pigment present on ventral tube and manubrium (Figure 14A–C).

Head: Antenna not annulated and 0.67–0.77 times length of body. Ratio of Ant. I–IV as 1.00/1.24–1.67/1.04–1.33/2.00–2.87. Distal part of Ant. IV with many sensory chaetae and normal ciliate chaetae, apical bulb bilobed (Figure 15A). Ant. III sense organ with two rods, two spiny guard sensilla, smooth blunt sens and ciliated chaetae (Figure 15B). Ant. II with 3–4 rods apically (Figure 15C). Prelabral and labral chaetae as 4/5, 5, 4, all smooth, a2 and b2 slightly shorter than middle ones (Figure 15D). Eyes 8 + 8, G and H smaller than others, interocular chaetae as p, r, t mes. Dorsal chaetotaxy of head with four antennal (An), five median (M) and eight sutural (S) mac (Figure 15E). Basal chaeta on maxillary outer lobe slightly thicker than apical one; sublobal plate with three smooth chaetae-like processes (Figure 15F). Lateral process (l. p.) of labial palp E differentiated with tip not reaching apex of papilla E (Figure 15G). Labial base with MReL_1_L_2_, e smooth, other ciliate, L_1_ rarely smooth, R 0.64–0.69 length of M (Figure 15H).

Thorax: Tergal ms formula on Th. II–Abd. V as 1, 0/1, 0, 1, 0, 0, sens as 2, 2/1, 2, 2, 2, 3 (Figure 16A, Figure 17, Figure 18 and Figure 19A). Th. II with four medio-medial (m1, m2, m2i, m2i2) mac, three medio-sublateral (m4, m4i, m4p), 34–47 posterior mac. Th. III with 41–54 mac (Figure 16A). Coxal macrochaetal formula as 3/4 + 1, 3/4 + 2 (Figure 16B–D). Trochanteral organ with 86–94 smooth chaetae (Figure 16E). Tenent hair clavate, 0.90–1.11 length of inner edge of unguis; unguis with 3–4 inner teeth, basal pair located at 0.36–0.42 distance from base of inner edge of unguis, unpaired tooth at 0.65–0.68 and 0.86 distance from base, respectively, most distal one very faint and usually absent; unguiculus lanceolate, outer edge slightly serrate (Figure 16F,G).

Abdomen: Range of Abd. IV length as 4.43–6.04 times as dorsal axial length of Abd. III. Abd. I with 11 (a1a, a1–3, m2i, m2–4, m4i, m4p, a5) mac. Abd. II with six (a2, a3, m3, m3e, m3ea, m3ep) central, one (m5) lateral mac. Abd. III with two (a2, m3) central, four (am6, pm6, m7a, p6) lateral mac (Figure 17). Abd. IV with two normal sens, 9 (rarely 12) anterior, six posterior and 17–20 lateral mac or mes (Figure 18). Abd. V with three sens (Figure 19A). Anterior face of ventral tube with 27–30 ciliate chaetae, 3+3 of them as mac, line connecting proximal (Pr) and external-distal (Ed) mac oblique to median furrow (Figure 19B); posterior face with five smooth chaetae apically (Figure 19C); lateral flap with 13–19 smooth and 4–8 ciliate chaetae (Figure 19D). Manubrial plate dorsally with 10–13 ciliate mac and three pseudopores (Figure 19E); ventrally with 26–33 ciliate chaetae on each side (Figure 19F). Dens with about 36 smooth inner spines (Figure 19G). Mucro bidentate with subapical tooth larger than apical one; tip of basal spine reaching apex of subapical tooth; distal smooth section of dens almost equal to mucro in length (Figure 19H,I).

Remarks

The new species can be distinguished from the other known species of genus *Homidia* by its colour pattern and it is similar to *H. oligoseta* Zhou, Huang & Ma, 2024 [22] and *H. quadriseta* Pan, 2018 [23] on the character. However, there are some differences between them in the chaetotaxy of Abd. IV and tenent hair. The detailed characteristic comparisons are listed in Table 6.

### 3.2. Neighbour-Joining Tree Analysis

Most sequenced individuals in the present study have a mean K2-P distance of COI sequences more than 14.0% (Figure 20, Table 7) and the number is greater than or comparable to the previous reported interspecific distances of 11.4% in the *Tomocerus nigrus* complex (Zhang et al., 2014) [24], 13.0% in *Plutomurus* (Barjadze et al., 2016) [25] and 16.2% in *Coecobrya* (Zhang et al., 2018) [26]. This indicates that the species delimitations are distinct among these species. However, the genetic distance between *H. laha* and *H. sauteri* is 0.0% and between *H. linhaiensis* and *H. tiantaiensis* is 0.3% (Table 7). 

#### 3.2.1. *H. laha* (Christiansen & Bellinger, 1992) and *H. sauteri* (Börner, 1906)

The genetic distance between *H. laha* and *H. sauteri* is 0.0, which indicates that they are the same species. The two species were discussed and synonymized by Ye et al. [27].

#### 3.2.2. *H. linhaiensis* Shi, Pan & Qi, 2009 and *H. tiantaiensis* Chen & Lin, 1998

The genetic distance between *H. linhaiensis* and *H. tiantaiensis* is only 0.3%, which is much less than the accepted barcoding gap. The dorsal chaetotaxy of the two species are almost the same and both Abd. IV are with m2 mac, which is rarely present in *Homidia* [28,29]. Other morphological differences are very slight. The dental spines and smooth setae on lateral flap varies greatly intraspecifically (Table 8). After consulting with Dr. Zhixiang Pan, we think the species *H. linhaiensis* Shi, Pan & Qi, 2009 is a new synonym of *H. tiantaiensis* Chen & Lin, 1998.

#### 3.2.3. *H. cingula* (Börner, 1906) and *H. nigrifascia* Ma & Pan, 2017

*H. cingula* was redescribed by Zhang et al. [30] based on the specimens from South Sulawesi and Jawa Timur of Indonesia and Yunnan of China. Its colour pattern is almost the same with that of *H. nigrifascia* [21], reported from Guizhou, a province bordering Yunnan. Their different dorsal chaetotaxy on body and great genetic distance in COI (26.3%) show they are different species (Table 9).

## 4. Discussion

The genus *Homidia* is well represented and widespread in China, especially eastern China. Because of poor research, among 28 provinces or autonomous regions of China, there are no reports of *Homidia* from Hainan, Heilongjiang, Hebei, Henan, Inner Mongolia, Liaoning, Ningxia, Qinghai, Shandong and Xinjiang (municipalities and special administrative regions are not included for their small areas). During the biodiversity survey of the Yintiaoling National Nature Reserve of Chongqing Municipality in 2024, we found *Homidia* was the dominant entomobryid genus and the numbers of individual and species were the greatest. This is the first report of *Homidia* from Chongqing and it suggests that many more species of *Homidia* may remain to be found in China.

## Figures and Tables

**Figure 1 insects-16-00974-f001:**
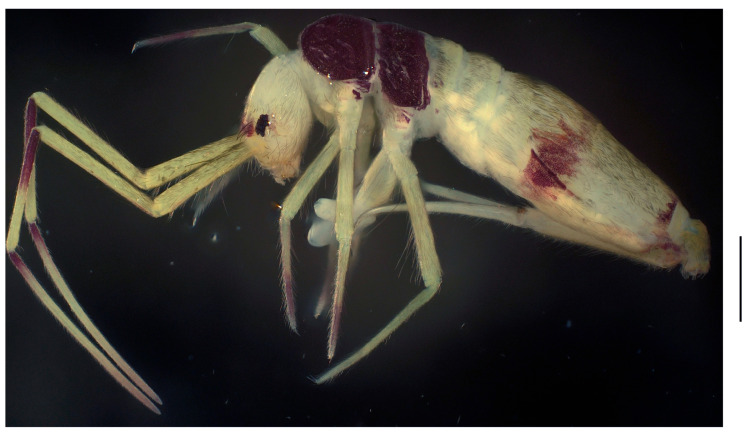
Habitus of *Homidia wuxiensis* sp. nov. (lateral view). Scale bar: 500 μm.

**Figure 2 insects-16-00974-f002:**
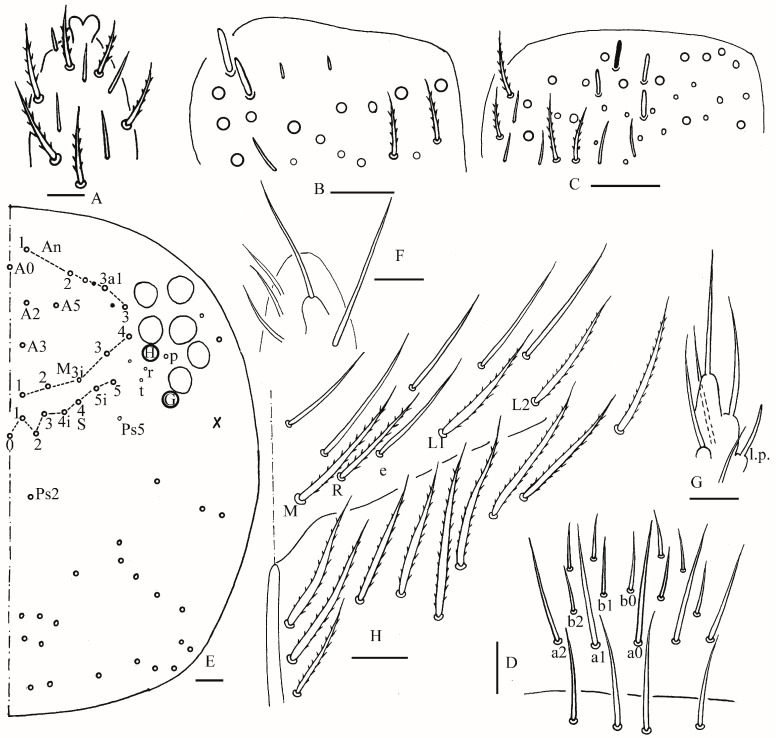
*Homidia wuxiensis* sp. nov. (**A**) apex of Ant. IV (dorsal view); (**B**) distal Ant. III (ventral view); (**C**) distal Ant. II (ventral view) (solid rod may absent); (**D**) prelabrum and labrum; (**E**) dorsal head (right side) (solid circle may absent); (**F**) maxillary palp and outer lobe (right side); (**G**) labial palp (right side); (**H**) labial and post-labial chaetotaxy (right side). Scale bars: 20 μm.

**Figure 3 insects-16-00974-f003:**
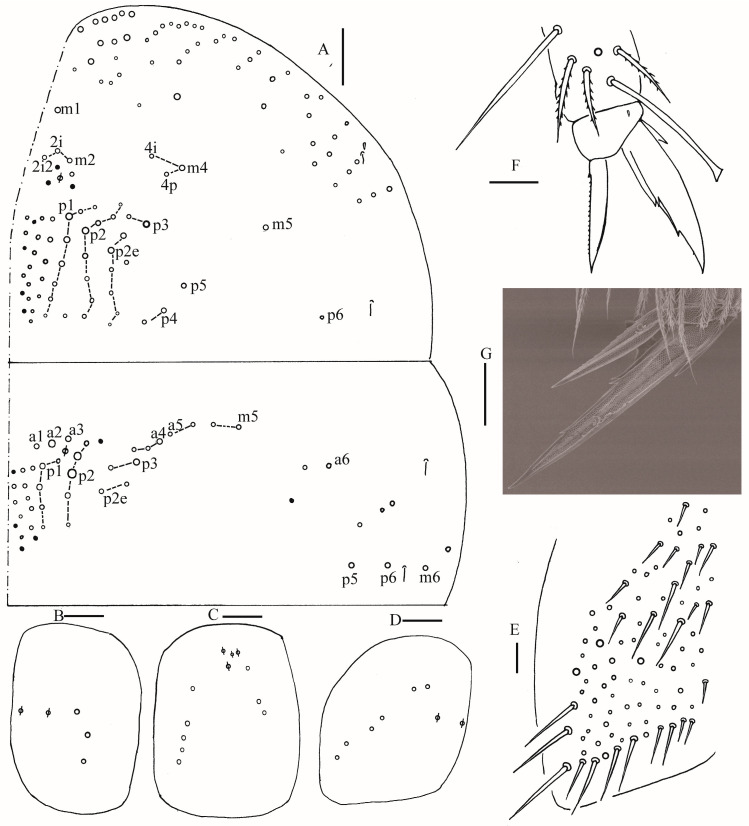
*Homidia wuxiensis* sp. nov. (**A**) chaetotaxy of Th. II–III (right side); (**B**–**D**) coxal chaetotaxy of fore, middle and hind leg; (**E**) trochanteral organ; (**F**) hind foot complex (lateral view); (**G**) SEM photomicrograph of hind foot complex (lateral view). Scale bars: (**A**) 50 μm; (**B**–**G**) 20 μm.

**Figure 4 insects-16-00974-f004:**
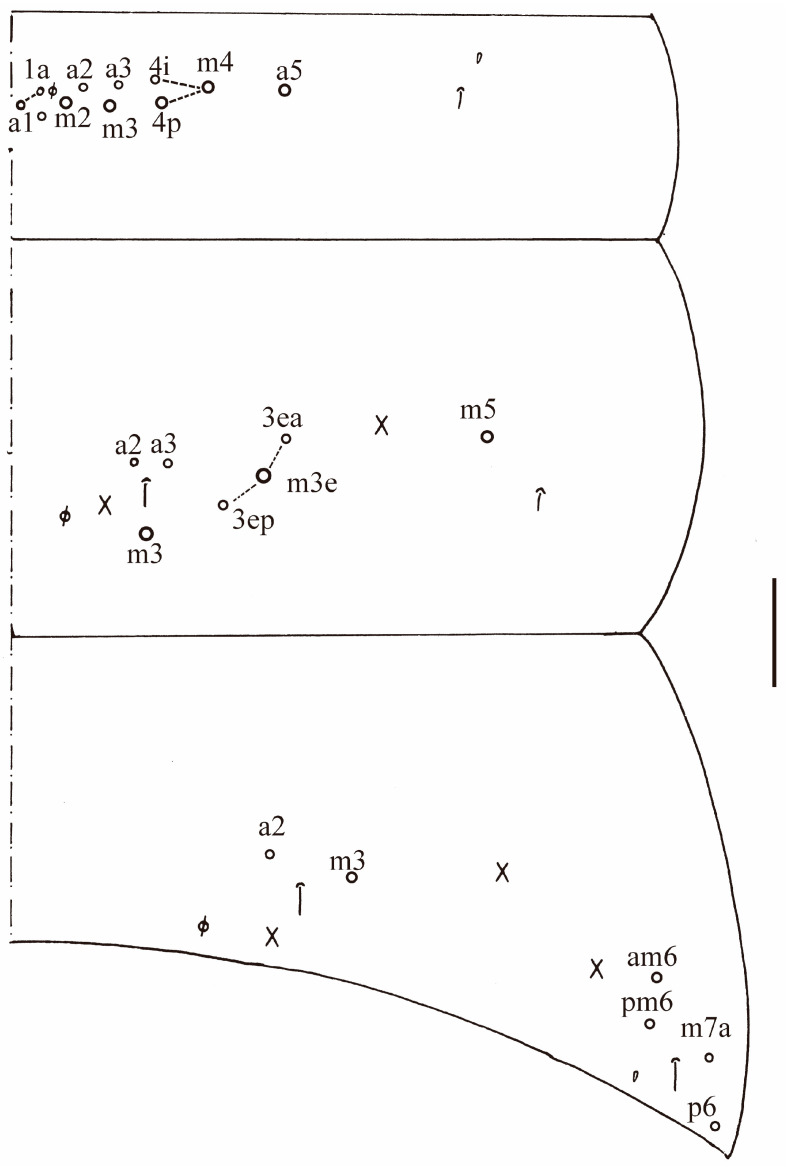
Chaetotaxy of Abd. I–III of *Homidia wuxiensis* sp. nov. (right side). Scale bar: 50 μm.

**Figure 5 insects-16-00974-f005:**
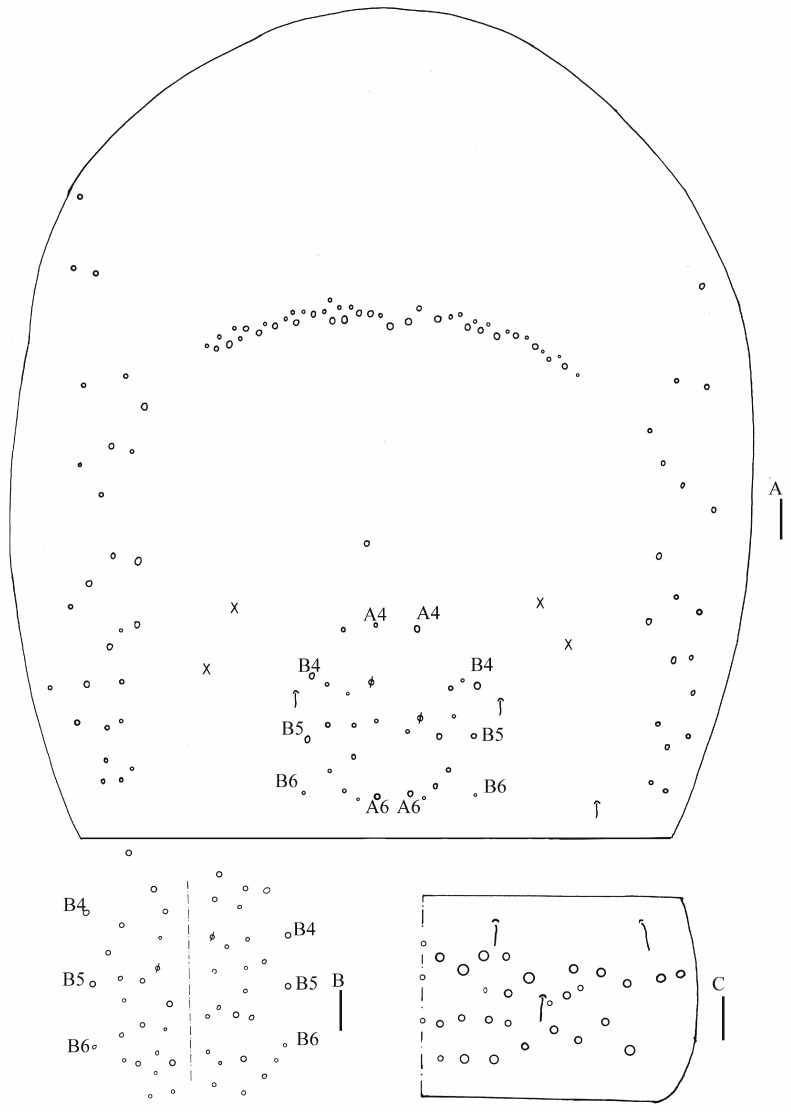
*Homidia wuxiensis* sp. nov. (**A**) chaetotaxy of Abd. IV (right side); (**B**) central chaetotaxy of Abd. IV posteriorly; (**C**) chaetotaxy of Abd. V (right side). Scale bars: (**A**,**B**) 50 μm; (**C**) 20 μm.

**Figure 6 insects-16-00974-f006:**
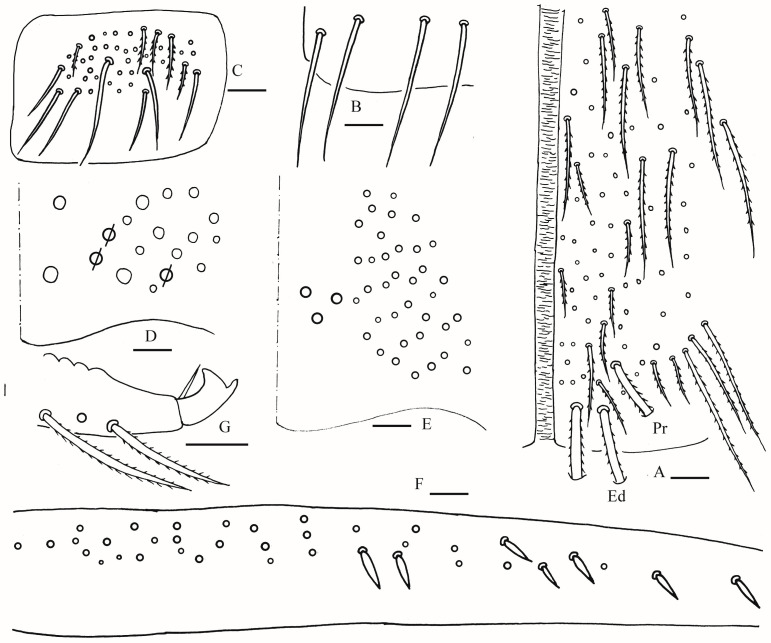
*Homidia wuxiensis* sp. nov. (**A**) anterior face of ventral tube; (**B**) smooth chaetae of posterior face of ventral tube apically; (**C**) lateral flap of ventral tube; (**D**) manubrial plaque (dorsal view); (**E**) ventro-apical part of manubrium; (**F**) proximal and median section of dens (circles also representing spines); (**G**) mucro (lateral view). Scale bars: 20 μm.

**Figure 7 insects-16-00974-f007:**
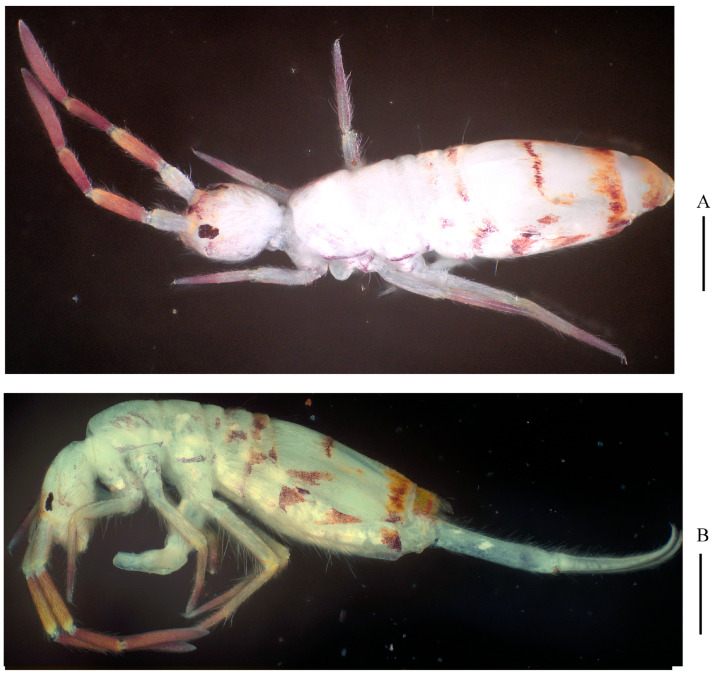
Habitus of *Homidia pseudochroma* sp. nov. (**A**) dorsal view.; (**B**) lateral view. Scale bars: 500 μm.

**Figure 8 insects-16-00974-f008:**
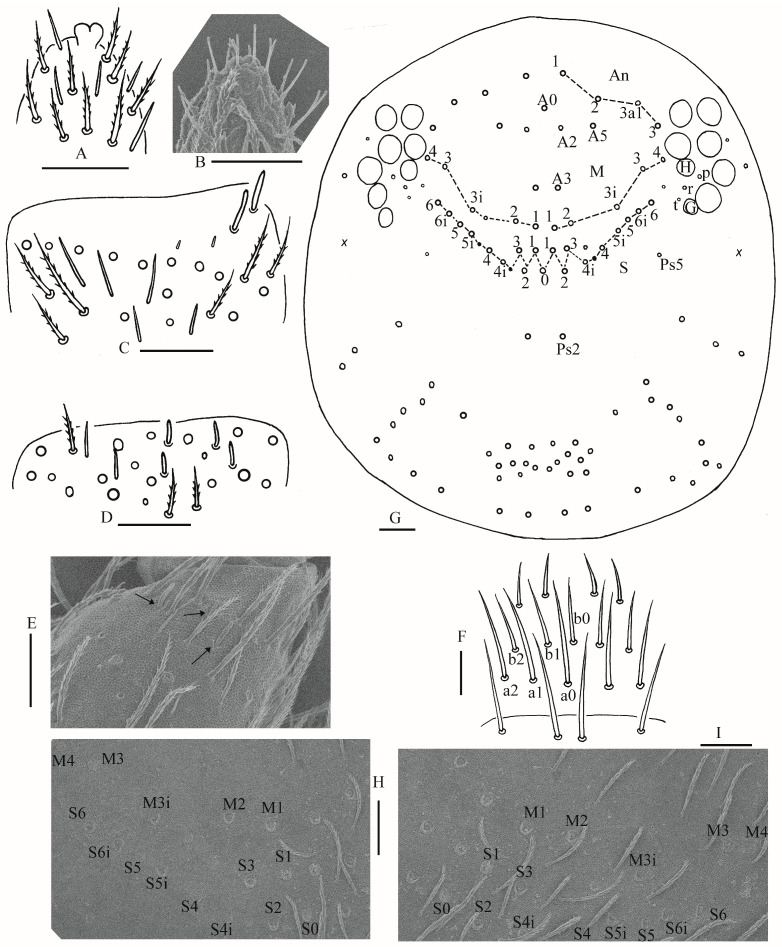
*Homidia pseudochroma* sp. nov. (**A**) apex of Ant. IV (dorsal view); (**B**) SEM photomicrograph of apex of Ant. IV (lateral view); (**C**) distal Ant. III (ventral view); (**D**) distal Ant. II (ventral view); (**E**) SEM photomicrograph of distal Ant. II (lateral view, black arrows showing rods); (**F**) prelabrum and labrum; (**G**) dorsal head; (**H**,**I**) SEM photomicrograph of dorsal head partially ((**H**) right side; (**I**) left side); Scale bars: 20 μm.

**Figure 9 insects-16-00974-f009:**
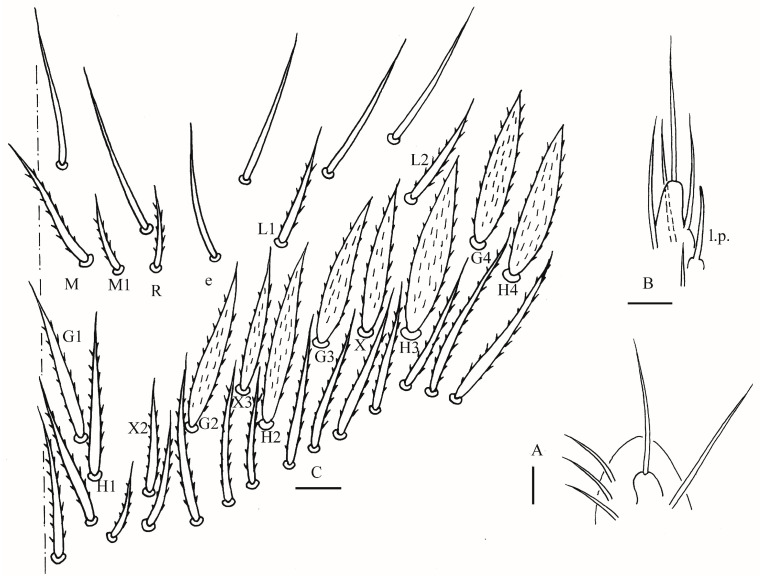
*Homidia pseudochroma* sp. nov. (**A**) maxillary palp and outer lobe (right side); (**B**) labial palp (right side); (**C**) labial and post-labial chaetotaxy (right side). Scale bars: 20 μm.

**Figure 10 insects-16-00974-f010:**
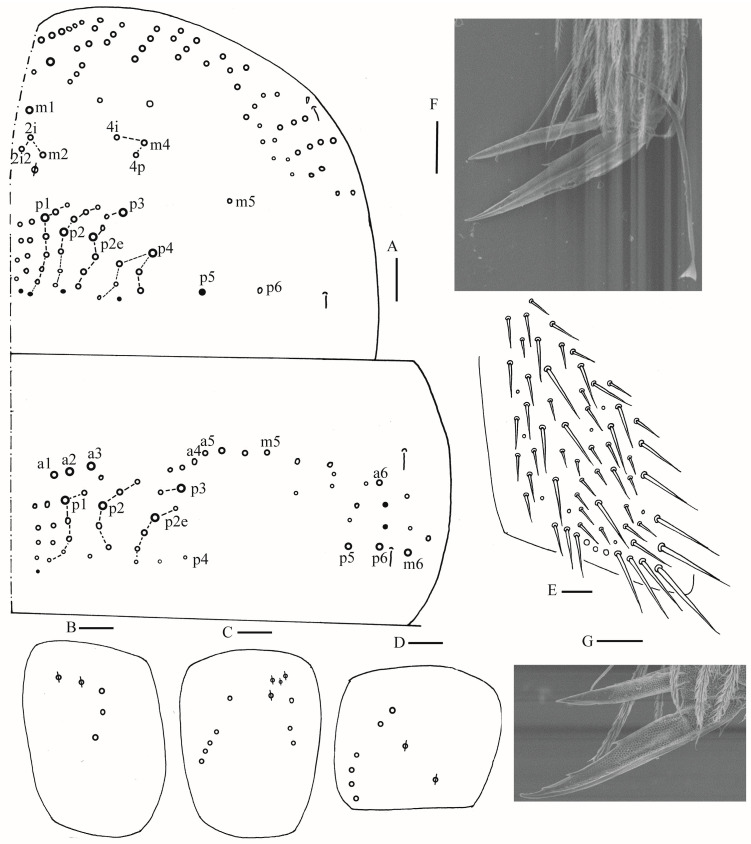
*Homidia pseudochroma* sp. nov. (**A**) chaetotaxy of Th. II–III (right side); (**B**–**D**) coxal chaetotaxy of fore, middle and hind leg; (**E**) trochanteral organ; (**F**,**G**) SEM photomicrograph of hind foot complex (lateral view). Scale bars: (**A**) 50 μm; (**B**–**G**) 20 μm.

**Figure 11 insects-16-00974-f011:**
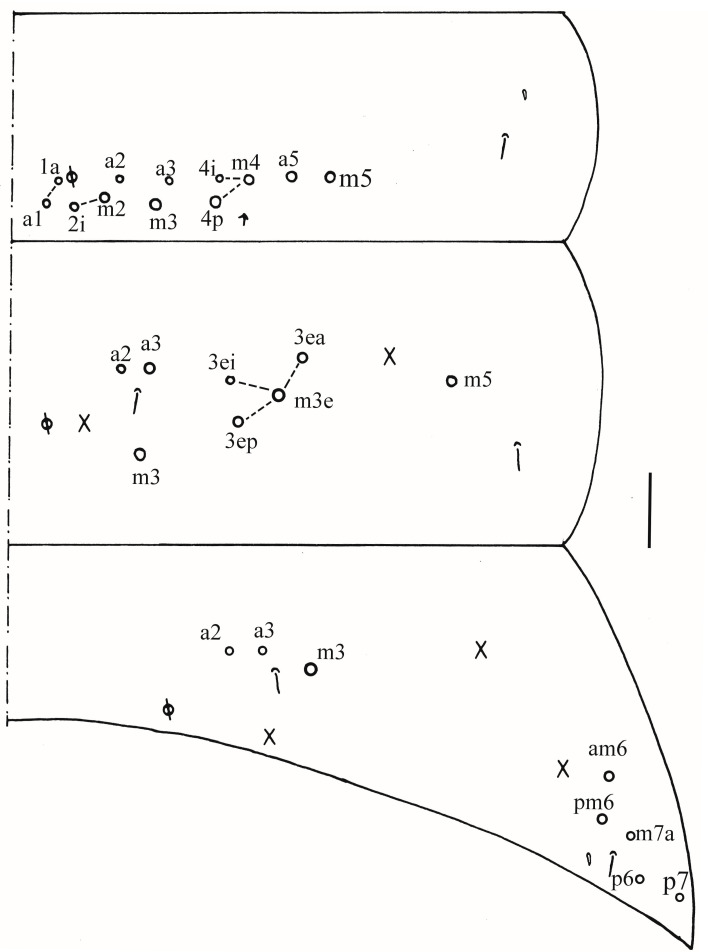
Chaetotaxy of Abd. I–III of *Homidia pseudochroma* sp. nov. (right side). Scale bar: 50 μm.

**Figure 12 insects-16-00974-f012:**
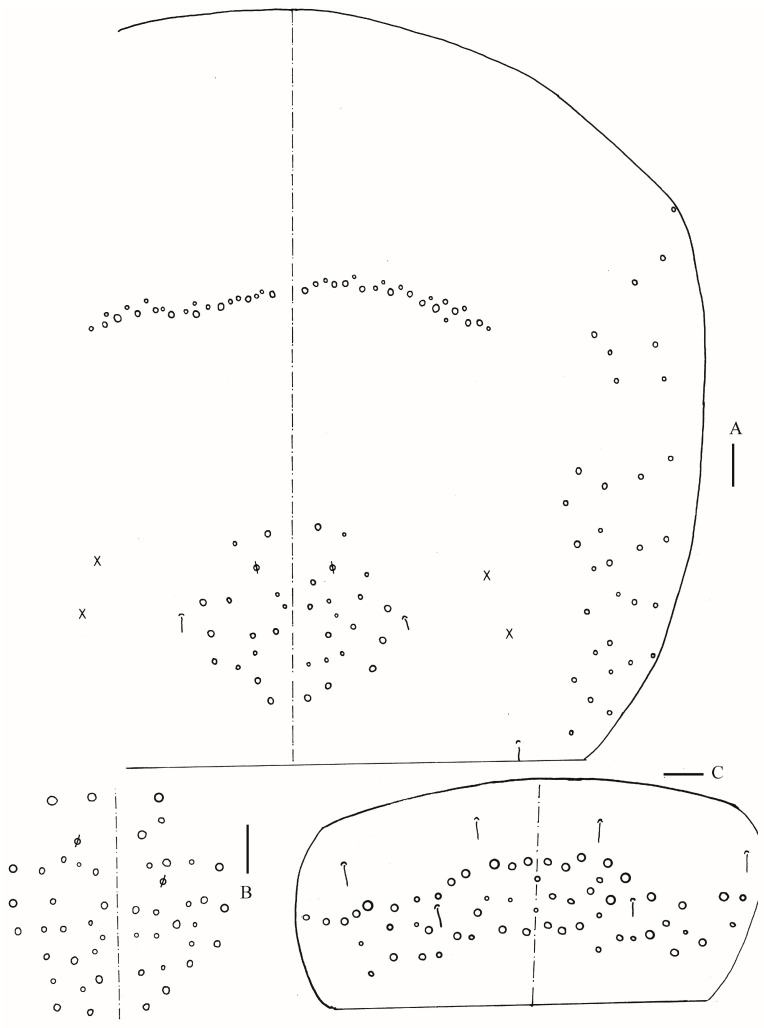
*Homidia pseudochroma* sp. nov. (**A**) chaetotaxy of Abd. IV (right side); (**B**) central chaetotaxy of Abd. IV posteriorly; (**C**) chaetotaxy of Abd. V. Scale bars: (**A**,**B**) 50 μm; (**C**) 20 μm.

**Figure 13 insects-16-00974-f013:**
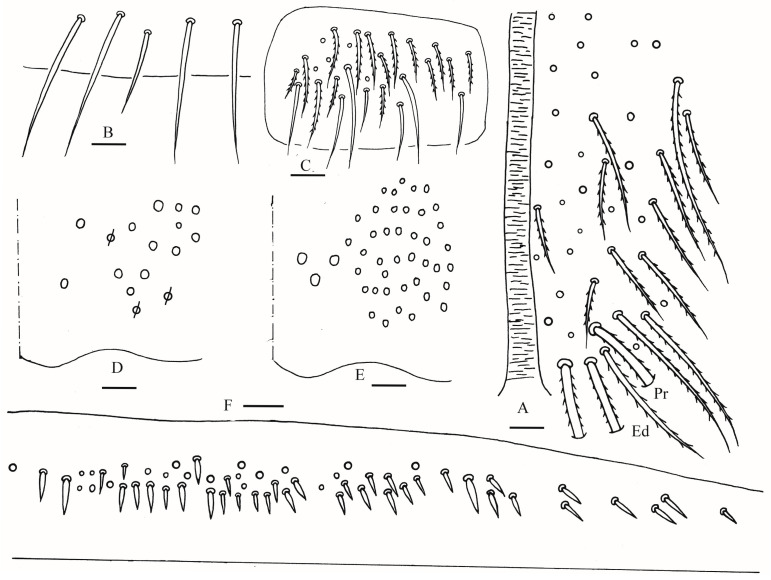
*Homidia pseudochroma* sp. nov. (**A**) anterior face of ventral tube; (**B**) smooth chaetae of posterior face of ventral tube apically; (**C**) lateral flap of ventral tube; (**D**) manubrial plaque (dorsal view); (**E**) ventro-apical part of manubrium; (**F**) proximal and median section of dens (circles also representing spines). Scale bars: 20 μm.

**Figure 14 insects-16-00974-f014:**
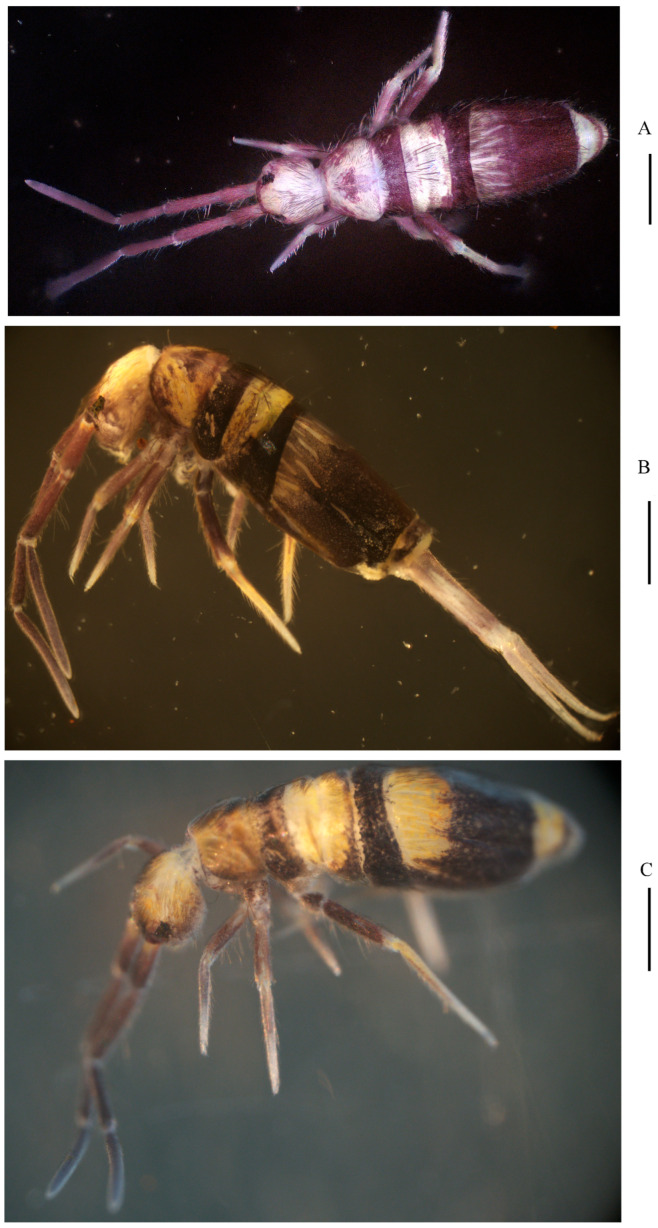
Habitus of *Homidia yangi* sp. nov. ((**A**) dorsal view; (**B**) lateral view; (**C**) dorsal view). Scale bars: 500 μm.

**Figure 15 insects-16-00974-f015:**
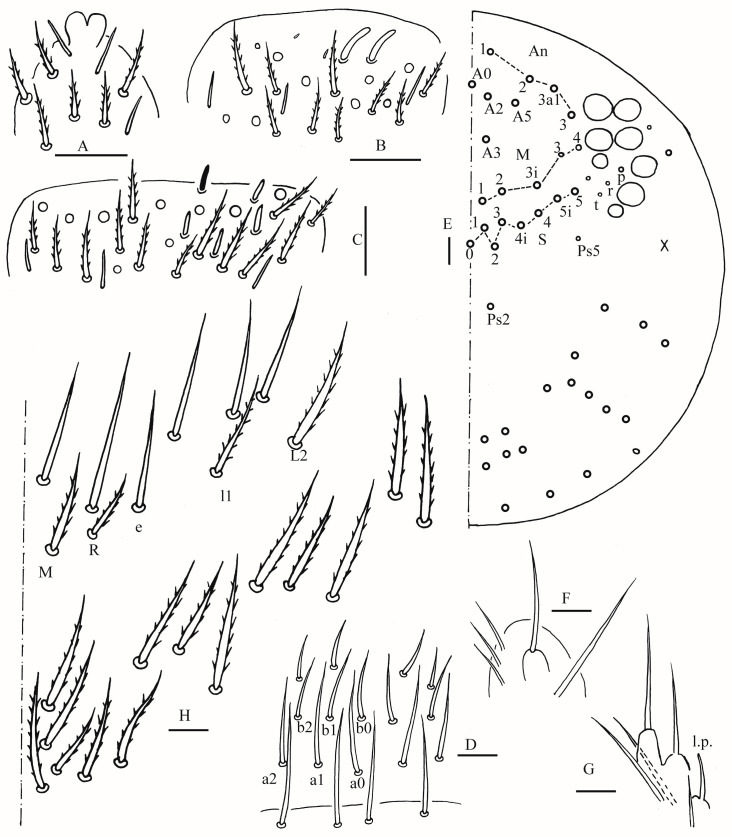
*Homidia yangi* sp. nov. (**A**) apex of Ant. IV (dorsal view); (**B**) distal Ant. III (ventral view); (**C**) distal Ant. II (ventral view); (**D**) prelabrum and labrum; (**E**) dorsal head (right side); (**F**) maxillary palp and outer lobe (right side); (**G**) labial palp (right side); (**H**) labial and post-labial chaetotaxy (right side). Scale bars: 20 μm.

**Figure 16 insects-16-00974-f016:**
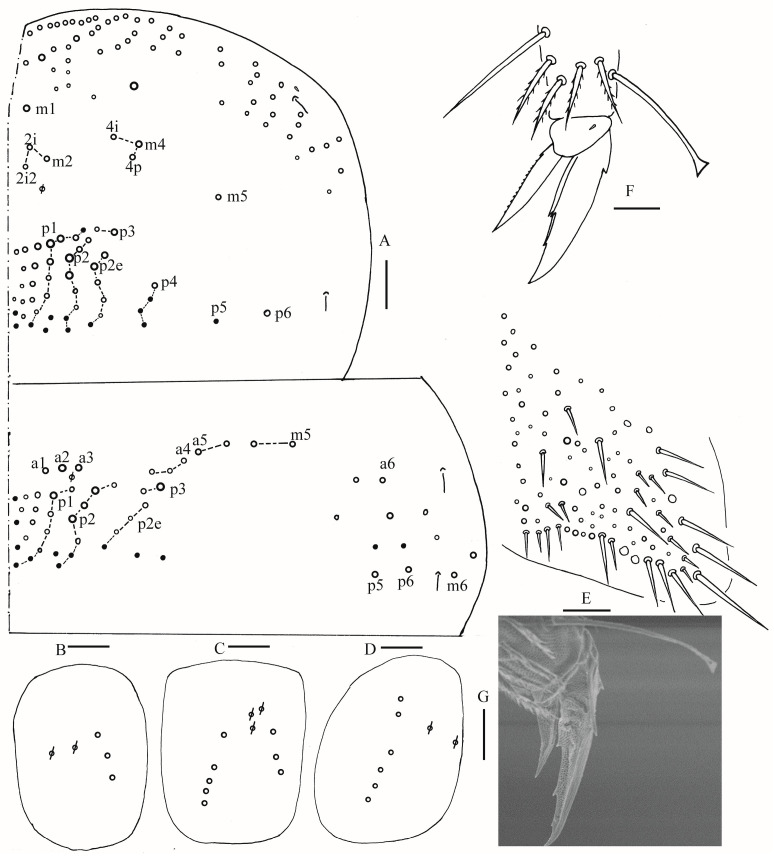
*Homidia yangi* sp. nov. (**A**) chaetotaxy of Th. II–III (right side); (**B**–**D**) coxal chaetotaxy of fore, middle and hind leg; (**E**) trochanteral organ; (**F**) hind foot complex (lateral view); (**G**) SEM photomicrograph of hind foot complex (lateral view). Scale bars: (**A**) 50 μm; (**B**–**G**) 20 μm.

**Figure 17 insects-16-00974-f017:**
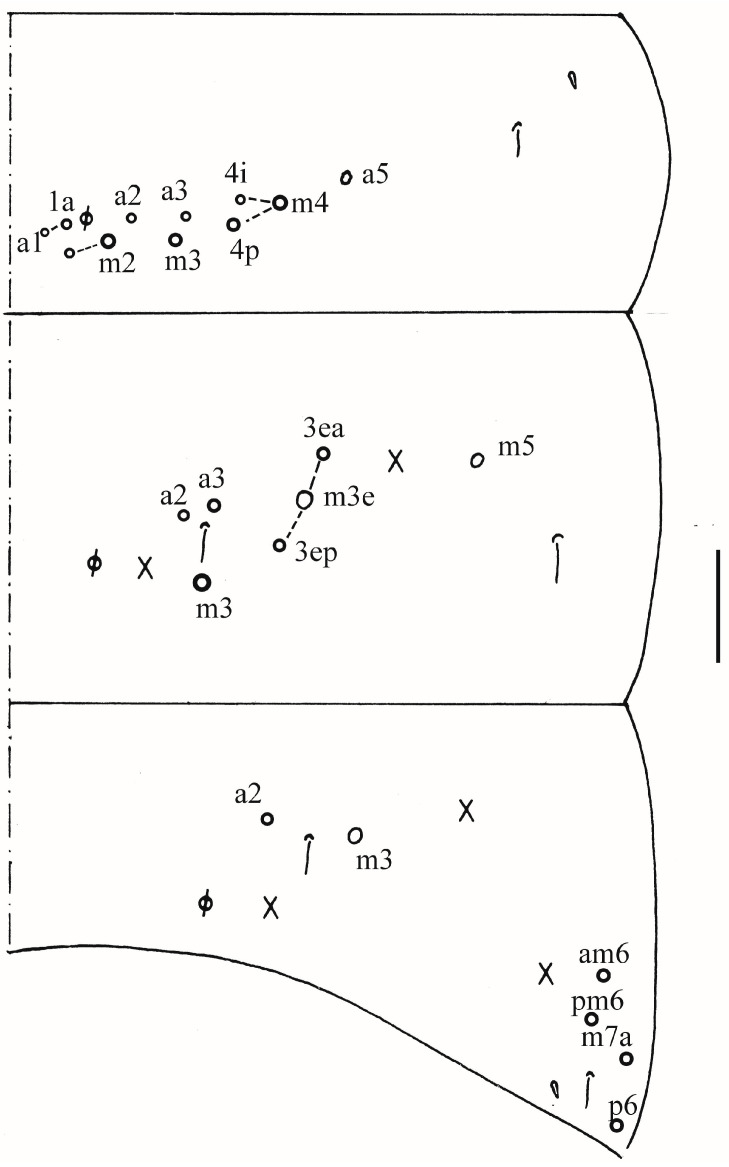
Chaetotaxy of Abd. I–III of *Homidia yangi* sp. nov. (right side). Scale bar: 50 μm.

**Figure 18 insects-16-00974-f018:**
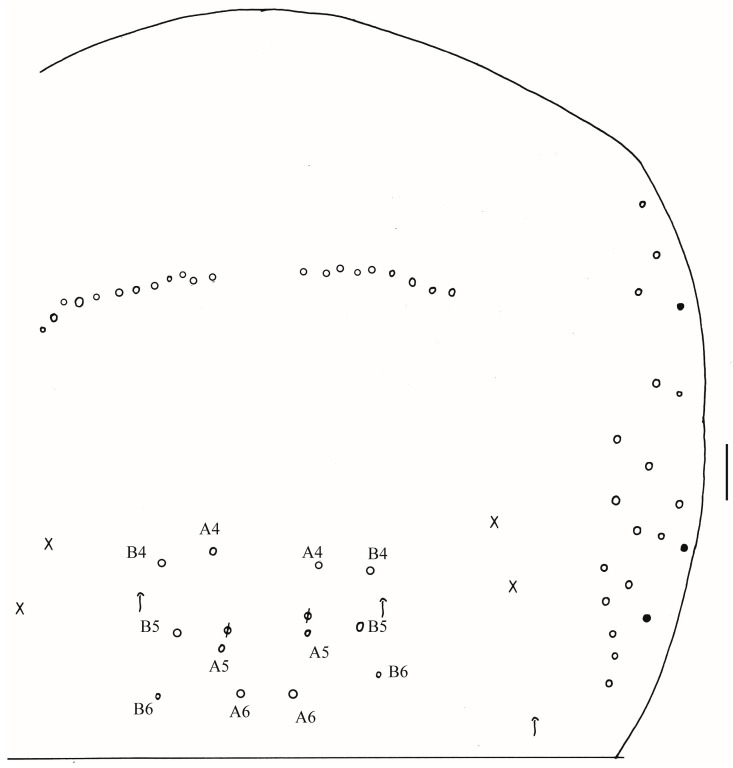
Chaetotaxy of Abd. IV of *Homidia yangi* sp. nov. (right side) Scale bar: 50 μm.

**Figure 19 insects-16-00974-f019:**
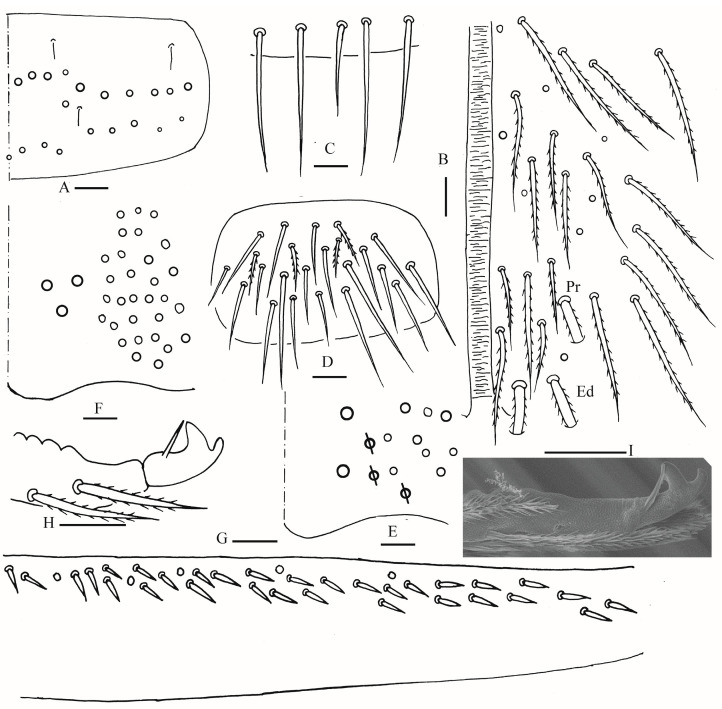
*Homidia yangi* sp. nov. (**A**) chaetotaxy of Abd. V; (**B**) anterior face of ventral tube; (**C**) smooth chaetae of posterior face of ventral tube apically; (**D**) lateral flap of ventral tube; (**E**) manubrial plaque (dorsal view); (**F**) ventro-apical part of manubrium; (**G**) proximal and median section of dens (circles also representing spines); (**H**) mucro (lateral view); (**I**) SEM photomicrograph of mucro (lateral view). Scale bars: 20 μm.

**Figure 20 insects-16-00974-f020:**
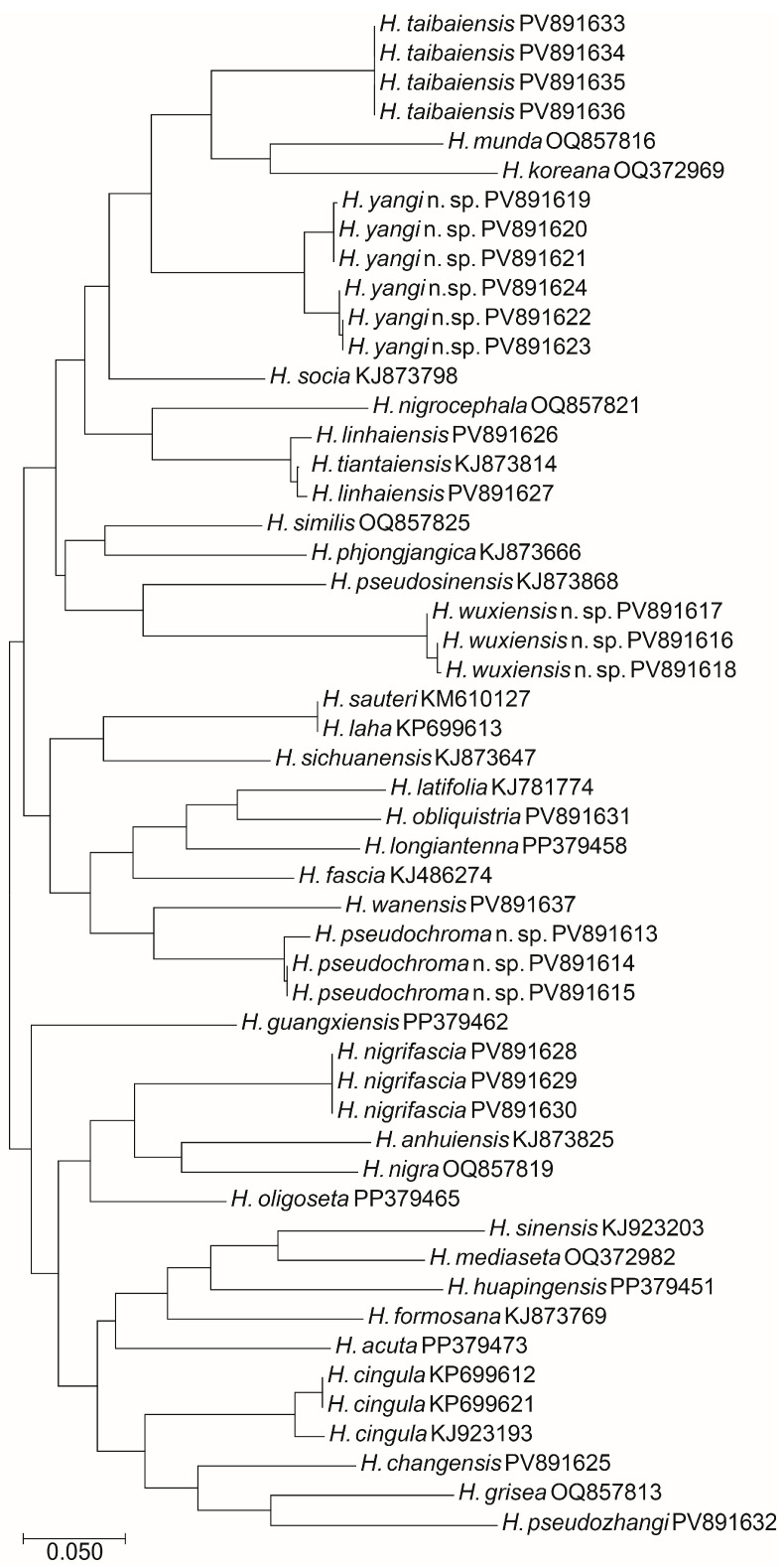
Neighbour-joining tree (using the K2-P model) of *Homidia* species based on COI sequences.

**Table 1 insects-16-00974-t001:** List of *Homidia* species from China *.

**Anhui Province (4)**	**Guangxi Zhuang Autonomous Region (7)**	**Jilin** **Province (6)**	**Shanxi Province (1)**
*H. anhuiensis* Li & Chen, 1997	*H. guangxiensis* Zhou, Huang & Ma, 2024	*H. hjesanica* Szeptycki, 1973	*H. sauteri* (Börner, 1909)
*H. qimenensis* Yi & Chen, 1999	*H. huapingensis* Zhou, Huang & Ma, 2024	*H. koreana* Lee & Lee, 1981	
*H. socia* Denis, 1929	*H. longiantenna* Zhou, Huang & Ma, 2024	*H. phjongjangica* Szeptycki, 1973	**Yunnan Province (3)**
*H. wanensis* Pan & Ma, 2021	*H. oligoseta* Zhou, Huang & Ma, 2024	*H. similis* Szeptycki, 1973	*H. cingula* (Börner, 1906)
	*H. qimenensis* Yi & Chen, 1999	*H. socia* Denis, 1929	*H. sauteri* (Börner, 1909)
**Beijing Municipality (1)**	*H. sichuanensis* Jia, Zhang & Jordana, 2010	*H. speciosa* Szeptycki, 1973	*H. sinensis* Denis, 1929
*H. sinensis* Denis, 1929	*H. socia* Denis, 1929		
		**Guizhou Province (4)**	**Zhejiang** **Province (22)**
**Fujian** **Province (7)**	**Hubei Province (1)**	*H. nigrifascia* Ma & Pan, 2017	*H. formosana* Uchida, 1943
*H. apigmenta* Shi, Pan & Zhang, 2010	*H. ziguiensis* Jia, Chen & Christiansen, 2003	*H. obliquistria* Ma & Pan, 2017	*H. hangzhouensis* Pan & Ma, 2021
*H. pseudosinensis* Shi & Pan, 2012		*H. qianensis* Jing & Ma, 2023	*H. hexaseta* Pan, Shi & Zhang, 2011
*H. qimenensis* Yi & Chen, 1999	**Hunan Province (1)**	*H. sichuanensis* Jia, Zhang & Jordana, 2010	*H. jordanai* Pan, Shi & Zhang, 2011
*H. similis* Szeptycki, 1973	*H. polyseta* Chen, 1998		*H. laha* (Christiansen & Bellinger, 1992)
*H. sinensis* Denis, 1929			*H. latifolia* Chen & Li, 1999
*H. socia* Denis, 1929	**Jiangsu Province (5)**	**Sichuan Province (2)**	*H. linhaiensis* Shi, Pan & Qi, 2009
*H. transitoria* Denis, 1929	*H. fascia* Wang & Chen, 2001	*H. emeiensis* Jia, Chen & Christiansen, 2004	*H. phjongjangica* Szeptycki, 1973
	*H. pentachaeta* Li & Christiansen, 1997	*H. sichuanensis* Jia, Zhang & Jordana, 2010	*H. pseudozhangi* Jing & Ma, 2023
**Gansu** **Province (1)**	*H. pseudofascia* Pan, Zhang & Li, 2015		*H. qimenensis* Yi & Chen, 1999
*H. maijiensis* Zhou & Ma, 2022	*H. qimenensis* Yi & Chen, 1999	**Taiwan Province (6)**	*H. quadrimaculata* Pan, 2015
	*H. socia* Denis, 1929	*H. formosana* Uchida, 1943	*H. quadriseta* Pan, 2018
**Guangdong** **Province (5)**		*H. linhaiensis* Shi, Pan & Qi, 2009	*H. sauteri* (Börner, 1909)
*H. chroma* Pan & Yang, 2019		*H. nigrocephala* Uchida, 1943	*H. similis* Szeptycki, 1973
*H. dianbaiensis* (Lin, 1985)	**Jiangxi Province (7)**	*H. sauteri* (Börner, 1909)	*H. sinensis* Denis, 1929
*H. huizhouensis* Zhou & Ma, 2022	*H. acuta* Jing & Ma, 2022	*H. socia* Denis, 1929	*H. socia* Denis, 1929
*H. leniseta* Pan & Yang, 2019	*H. changensis* Jing & Ma, 2022	*H. taibaiensis* Yuan & Pan, 2013	*H. tiantaiensis* Chen & Lin, 1998
*H. sichuanensis* Jia, Zhang & Jordana, 2010	*H. hexchaeta* Zhou & Ma, 2022		*H. triangulimacula* Pan & Shi, 2015
	*H. leei* Chen & Li, 1997	**Xizang Autonomous Region (4)**	*H. unichaeta* Pan, Shi & Zhang, 2010
**Shaanxi** **Province (3)**	*H. linhaiensis* Shi, Pan & Qi, 2009	*H. breviseta* Pan, 2022	*H. xianjuensis* Wu & Pan, 2016
*H. huashanensis* Jia, Chen & Christiansen, 2005	*H. socia* Denis, 1929	*H. sichuanensis* Jia, Zhang & Jordana, 2010	*H. yandangensis* Pan, 2015
*H. mediofascia* Shi, Pan & Bai, 2009	*H. pseudozhangi* Jing & Ma, 2023	*H. sinensis* Denis, 1929	*H. zhangi* Pan & Shi, 2012
*H. taibaiensis* Yuan & Pan, 2013		*H. tibetensis* Chen & Zhong, 1998	

* The bolds representing province or autonomous region of China.

**Table 2 insects-16-00974-t002:** List of *Homidia* species from other regions except China *.

**Korean Peninsula (16)**	**Japan (14)**	**Vietnam (7)**
*H. chosonica* Szeptycki, 1973	*H. allospila* (Börner, 1909)	*H. glassa* Nguyen, 2001
*H. flavonigra* Szeptycki, 1973	*H. amethystinoides* Jordana & Baquero, 2010	*H. lakhanpurensis* Baquero & Jordana, 2015
*H. grisea* Lee & Lee, 1981	*H. chrysothrix* Yosii, 1942	*H. multidentata* Nguyen, 2005
*H. heugsanica* Lee & Park, 1984	*H. cingula* (Börner, 1906)	*H. sinensis* Denis, 1929
*H. hjesanica* Szeptycki, 1973	*H. fujiyamai* Uchida, 1954	*H. socia* Denis, 1929
*H. koreana* Lee & Lee, 1981	*H. munda* Yosii, 1956	*H. subcingula* Denis, 1948
*H. mediaseta* Lee & Lee, 1981	*H. nigrocephala* Uchida, 1943	*H. unichaeta* Pan, Shi & Zhang, 2010
*H. minuta* Kim & Lee, 1995	*H. rosannae* Jordana & Baquero, 2010	
*H. munda* Yosii, 1956	*H. sauteri* (Börner, 1909)	**The United States of America (6)**
*H. nigra* Lee & Lee, 1981	*H. sinensis* Denis, 1929	*H. haikea* (Christiansen & Bellinger, 1992)
*H. phjongjangica* Szeptycki, 1973	*H. socia* Denis, 1929	*H. hihiu* (Christiansen & Bellinger, 1992)
*H. pseudoformosana* Kang & Park, 2012	*H. sotoi* Jordana & Baquero, 2010	*H. insularis* (Carpenter, 1904)
*H. pseudokoreana* Lee & Park, 2024	*H. subcingula* Denis, 1948	*H. laha* (Christiansen & Bellinger, 1992)
*H. sauteri* (Börner, 1909)	*H. yoshiii* Jordana & Baquero, 2010	*H. sauteri* (Börner, 1909)
*H. similis* Szeptycki, 1973		*H. socia* Denis, 1929
*H. speciosa* Szeptycki, 1973		
	**India (2)**	**Thailand (2)**
**Bangladesh, Indonesia, Malaya, Singapore (1)**	*H. cingula* (Börner, 1906)	*H. cingula* (Börner, 1906)
*H. cingula* (Börner, 1906)	*H. lakhanpurensis* Baquero & Jordana 2015	*H. subcingula* Denis, 1948

* The bolds representing country or region.

**Table 3 insects-16-00974-t003:** Species information and COI GenBank accession numbers.

Species	Voucher	GenBank	Data
*H. acuta* Jing & Ma, 2022	2902	PP379473	GenBank
*H. anhuiensis* Li & Chen, 1997	AHLA_1	KJ873825	GenBank
*H. changensis* Jing & Ma, 2022	1243-1	PV891625	This study
*H. cingula* (Börner, 1906)	14YN2_1	KP699612	GenBank
*H. cingula* (Börner, 1906)	14YN3-2	KP699621	GenBank
*H. cingula* (Börner, 1906)	JAVA05CV03	KJ923193	GenBank
*H. fascia* Wang & Chen, 2001	4399_1	KJ486274	GenBank
*H. formosana* Uchida, 1943	HNCS	KJ873769	GenBank
*H. grisea* Lee & Lee, 1981	HG_117-1_2	OQ857813	GenBank
*H. guangxiensis* Zhou, Huang & Ma, 2024	8302	PP379462	GenBank
*H. huapingensis* Zhou, Huang & Ma, 2024	7402	PP379451	GenBank
*H. koreana* Lee & Lee, 1981	HK_111-1	OQ372969	GenBank
*H. laha* (Christiansen & Bellinger, 1992)	14YN27_1	KP699613	GenBank
*H. latifolia* Chen & Li, 1999	HNCS_2	KJ781774	GenBank
*H. linhaiensis* Shi, Pan & Qi, 2009	1241-1	PV891626	This study
*H. linhaiensis* Shi, Pan & Qi, 2009	m1246-1	PV891627	This study
*H. longiantenna* Zhou, Huang & Ma, 2024	8103	PP379458	GenBank
*H. mediaseta* Lee & Lee, 1981	HM_114-1	OQ372982	GenBank
*H. munda* Yosii, 1956	HMU_116_3	OQ857816	GenBank
*H. nigra* Lee & Lee, 1981	HNG_131	OQ857819	GenBank
*H. nigrifascia* Ma & Pan, 2017	m1128-1	PV891628	This study
*H. nigrifascia* Ma & Pan, 2017	m1128-2	PV891629	This study
*H. nigrifascia* Ma & Pan, 2017	m1149-3	PV891630	This study
*H. nigrocephala* Uchida, 1943	HNCH_122-2	OQ857821	GenBank
*H. obliquistria* Ma & Pan, 2017	m1246-2	PV891631	This study
*H. oligoseta* Zhou, Huang & Ma, 2024	8306	PP379465	GenBank
*H. phjongjangica* Szeptycki, 1973	XB_1	KJ873666	GenBank
*H. pseudochroma* sp. nov.	m1306-11	PV891613	This study
*H. pseudochroma* sp. nov.	m1306-12	PV891614	This study
*H. pseudochroma* sp. nov.	m1306-13	PV891615	This study
*H. pseudosinensis* Shi & Pan, 2012	FJFZ_1	KJ873868	GenBank
*H. pseudozhangi* Jing & Ma, 2023	1235-1	PV891632	This study
*H. sauteri* (Börner, 1906)	12-19-Hsa-1	KM610127	GenBank
*H. sichuanensis* Jia, Zhang & Jordana, 2010	1	KJ873647	GenBank
*H. similis* Szeptycki, 1973	HS_5-3_167	OQ857825	GenBank
*H. sinensis* Denis, 1929	4017A11	KJ923203	GenBank
*H. socia* Denis, 1929	HBHS_1	KJ873798	GenBank
*H. taibaiensis* Yuan & Pan, 2013	m1306-14	PV891633	This study
*H. taibaiensis* Yuan & Pan, 2013	m1318-1	PV891634	This study
*H. taibaiensis* Yuan & Pan, 2013	m1306-19	PV891635	This study
*H. taibaiensis* Yuan & Pan, 2013	m1306-20	PV891636	This study
*H. tiantaiensis* Chen & Lin, 1998	AHLA_1	KJ873814	GenBank
*H. wanensis* Pan & Ma, 2021	m1189-2	PV891637	This study
*H. wuxiensis* sp. nov.	m1317-5	PV891616	This study
*H. wuxiensis* sp. nov.	m1317-6	PV891617	This study
*H. wuxiensis* sp. nov.	m1317-7	PV891618	This study
*H. yangi* sp. nov.	m1317-2	PV891619	This study
*H. yangi* sp. nov.	m1317-3	PV891620	This study
*H. yangi* sp. nov.	m1317-4	PV891621	This study
*H. yangi* sp. nov.	m1321-25	PV891622	This study
*H. yangi* sp. nov.	m1321-26	PV891623	This study
*H. yangi* sp. nov.	m1321-27	PV891624	This study

**Table 5 insects-16-00974-t005:** Main differences among the new species and similar species of *Homidia*.

Characters	*H. pseudochroma* sp. nov.	*H. chroma*	*H. obliquistria*	*H. pentachaeta*
S mac on head	10	8	8	8
Mac on Abd. I	12	11	11–12	10
M3ei mac on Abd. II	present	absent	absent	absent
Central mac on Abd. III	3	2	2	3
Lateral mac on Abd. III	5	4	4	5
Anterior mac or mes on Abd. IV	21–22	9–12	15–24	10–13
Central mac or mes on Abd. IV	14–18	6	13–33	12–14
Labial chaetotaxy	MM_1_ReL_1_L_2_	MRel_1_L_2_	MM_1_ReL_1_L_2_	MRel_1_L_2_
Expanded post-labial chaetae	present	absent	present	absent
A medial longitudinal strip on Th. II–III	absent	absent	present	absent

**Table 6 insects-16-00974-t006:** Main differences among the new species and similar species of *Homidia*.

Characters	*H. yangi* sp. nov.	*H. oligoseta*	*H. quadriseta*
A transverse brown band on Th. III	present	present	absent
A transverse brown band on Abd. II	absent	present	absent
Tenent hair	clavate	pointed	clavate
Central mac on Abd. IV anteriorly	9	3–4	2
Central mac on Abd. IV posteriorly	6	4 (5)	3

**Table 7 insects-16-00974-t007:** The genetic distance (mean K2-P divergence) within and between species in the study *.

**Species**	** *acu* **	** *anh* **	** *cha* **	** *cin* **	** *fas* **	** *for* **	** *gri* **	** *gua* **	** *hua* **	** *kor* **	** *lah* **	** *lat* **	** *lin* **	** *lon* **	** *med* **	** *mun* **	** *nia* **	** *nig* **
** *acu* **																		
** *anh* **	28.7																	
** *cha* **	24.0	29.9																
** *cin* **	22.3	28.2	18.9															
** *fas* **	29.7	31.8	30.9	29.2														
** *for* **	22.6	30.3	25.6	23.9	31.3													
** *gri* **	28.9	34.8	20.3	22.5	35.8	30.5												
** *gua* **	24.7	26.8	25.9	24.2	25.2	26.3	30.8											
** *hua* **	26.5	34.2	29.5	27.8	35.2	23.1	34.4	30.2										
** *kor* **	39.7	41.8	40.9	39.2	36.6	41.3	45.8	35.2	45.2									
** *lah* **	30.8	32.9	32.0	30.3	25.1	32.4	36.9	26.3	36.3	37.7								
** *lat* **	34.2	36.3	35.4	33.7	20.3	35.8	40.3	29.7	39.7	41.1	29.6							
** *lin* **	30.5	32.6	31.7	30.0	27.4	32.1	36.6	26.0	36.0	31.4	28.5	31.9						
** *lon* **	32.9	35.0	34.1	32.4	19.0	34.5	39.0	28.4	38.4	39.8	28.3	18.3	30.6					
** *med* **	25.6	33.3	28.6	26.9	34.3	22.2	33.5	29.3	21.9	44.3	35.4	38.8	35.1	37.5				
** *mun* **	37.1	39.2	38.3	36.6	34.0	38.7	43.2	32.6	42.6	19.6	35.1	38.5	28.8	37.2	41.7			
** *nia* **	28.0	17.9	29.2	27.5	31.1	29.6	34.1	26.1	33.5	41.1	32.2	35.6	31.9	34.3	32.6	38.5		
** *nig* **	26.8	21.3	28.0	26.3	29.9	28.4	32.9	24.9	32.3	39.9	31.0	34.4	30.7	33.1	31.4	37.3	20.6	
** *nio* **	33.3	35.4	34.5	32.8	30.2	34.9	39.4	28.8	38.8	34.2	31.3	34.7	17.4	33.4	37.9	31.6	34.7	33.5
** *obl* **	33.9	36.0	35.1	33.4	20.0	35.5	40.0	29.4	39.4	40.8	29.3	14.3	31.6	18.0	38.5	38.2	35.3	34.1
** *oli* **	21.5	20.4	22.7	21.0	24.6	23.1	27.6	19.6	27.0	34.6	25.7	29.1	25.4	27.8	26.1	32.0	19.7	18.5
** *phj* **	30.3	32.4	31.5	29.8	27.2	31.9	36.4	25.8	35.8	34.0	28.3	31.7	24.8	30.4	34.9	31.4	31.7	30.5
** *psc* **	29.2	31.3	30.4	28.7	19.5	30.8	35.3	24.7	34.7	36.1	24.6	24.0	26.9	22.7	33.8	33.5	30.6	29.4
** *pss* **	31.2	33.3	32.4	30.7	28.1	32.8	37.3	26.7	36.7	34.9	29.2	32.6	25.7	31.3	35.8	32.3	32.6	31.4
** *psz* **	31.0	36.9	22.4	24.6	37.9	32.6	20.1	32.9	36.5	47.9	39.0	42.4	38.7	41.1	35.6	45.3	36.2	35.0
** *sau* **	30.8	32.9	32.0	30.3	25.1	32.4	36.9	26.3	36.3	37.7	0.0	29.6	28.5	28.3	35.4	35.1	32.2	31.0
** *sic* **	28.5	30.6	29.7	28.0	22.8	30.1	34.6	24.0	34.0	35.4	18.7	27.3	26.2	26.0	33.1	32.8	29.9	28.7
** *sim* **	28.1	30.2	29.3	27.6	25.0	29.7	34.2	23.6	33.6	31.8	26.1	29.5	22.6	28.2	32.7	29.2	29.5	28.3
** *sin* **	28.5	36.2	31.5	29.8	37.2	25.1	36.4	32.2	24.8	47.2	38.3	41.7	38.0	40.4	17.3	44.6	35.5	34.3
** *soc* **	28.2	30.3	29.4	27.7	25.1	29.8	34.3	23.7	33.7	26.7	26.2	29.6	19.9	28.3	32.8	24.1	29.6	28.4
** *tai* **	33.7	35.8	34.9	33.2	30.6	35.3	39.8	29.2	39.2	22.0	31.7	35.1	25.4	33.8	38.3	19.4	35.1	33.9
** *tia* **	29.5	31.6	30.7	29.0	26.4	31.1	35.6	25.0	35.0	30.4	27.5	30.9	0.3	29.6	34.1	27.8	30.9	29.7
** *wan* **	32.0	34.1	33.2	31.5	22.3	33.6	38.1	27.5	37.5	38.9	27.4	26.8	29.7	25.5	36.6	36.3	33.4	32.2
** *wux* **	36.2	38.3	37.4	35.7	33.1	37.8	42.3	31.7	41.7	39.9	34.2	37.6	30.7	36.3	40.8	37.3	37.6	36.4
** *yan* **	31.6	33.7	32.8	31.1	28.5	33.2	37.7	27.1	37.1	25.9	29.6	33.0	23.3	31.7	36.2	23.3	33.0	31.8
**Species**	** *nio* **	** *obl* **	** *o* ** ** *li* **	** *phj* **	** *psc* **	** *pss* **	** *psz* **	** *sau* **	** *sic* **	** *sim* **	** *sin* **	** *soc* **	** *tai* **	** *tia* **	** *wan* **	** *wux* **	** *yan* **	
** *nio* **																		
** *obl* **	34.4																	
** *oli* **	28.2	28.8																
** *phj* **	27.6	31.4	25.2															
** *psc* **	29.7	23.7	24.1	26.7														
** *pss* **	28.5	32.3	26.1	24.5	27.6													
** *psz* **	41.5	42.1	29.7	38.5	37.4	39.4												
** *sau* **	31.3	29.3	25.7	28.3	24.6	29.2	39.0											
** *sic* **	29.0	27.0	23.4	26.0	22.3	26.9	36.7	18.7										
** *sim* **	25.4	29.2	23.0	17.6	24.5	22.3	36.3	26.1	23.8									
** *sin* **	40.8	41.4	29.0	37.8	36.7	38.7	38.5	38.3	36.0	35.6								
** *soc* **	22.7	29.3	23.1	22.5	24.6	23.4	36.4	26.2	23.9	20.3	35.7							
** *tai* **	28.2	34.8	28.6	28.0	30.1	28.9	41.9	31.7	29.4	25.8	41.2	20.7						
** *tia* **	17.4	30.6	24.4	23.8	25.9	24.7	37.7	27.5	25.2	21.6	37.0	18.9	24.4					
** *wan* **	32.5	26.5	26.9	29.5	15.6	30.4	40.2	27.4	25.1	27.3	39.5	27.4	32.9	28.7				
** *wux* **	33.5	37.3	31.1	29.5	32.6	22.8	44.4	34.2	31.9	27.3	43.7	28.4	33.9	29.7	35.4			
** *yan* **	26.1	32.7	26.5	25.9	28.0	26.8	39.8	29.6	27.3	23.7	39.1	18.6	19.9	22.3	30.8	31.8		

* The bolds are species abbreviations: *acu, H. acuta; anh, H. anhuiensis; cha, changensis; cin, H. cingula; fas, H. fascia; for, H. formosana; gri, H. grisea; gua, H. guangxiensis; hua, H. huapingensis; kor, H. koreana; lah, H. laha; lat, H. latifolia; lin, H. linhaiensis; lon, H. longiantenna; med, H. mediaseta; mun, H. munda; nia, H. nigra; nig, H. nigrifascia; nio, H. nigrocephala; obl, H. obliquistria; oli, H. oligoseta; phj, H. phjongjangica; psc, H. pseudochroma* sp. nov.; *pss, H. pseudosinensis; psz, H. pseudozhangi; sau, H. sauteri; sic, H. sichuanensis; sim, H. similis; sin, H. sinensis; soc, H. socia; tai, H. taibaiensis; tia, H. tiantaiensis; wan, H. wanensis; wux, H. wuxiensis* sp. nov.; *yan, H. yangi* sp. nov.

**Table 8 insects-16-00974-t008:** Main comparison between *H. linhaiensis* and *H. tiantaiensis*.

Characters	*H. linhaiensis*	*H. tiantaiensis*
A pair of dark spots on Th. III	present	present
A pair of dark spots on Abd. III	present	absent or present
Labial chaetotaxy	MRel_1_L_2_	MRel_1_L_2_
Dorsal chaetotaxy of head	7 An, 5 M, 9 S mac	5 An, 4 M, 9 S mac
Medio-medial mac on Th. II	4	4
Medio-sublateral mac on Th. II	4	3
Posterior mac on Th. II	26–29	35
Mac on Abd. I	10 (11)	10
Central mac on Abd. II	6	6
Central mac on Abd. III	2	2
Lateral mac on Abd. III	5	4
A2 mac on Abd. IV	present	present
Central mac on Abd. IV anteriorly	10–13	8–12
Central mac on Abd. IV posteriorly	10–16	13–16
Ungual inner teeth	4	3
Smooth setae on lateral flap	9	5
Dental spines	6–16	29–36

**Table 9 insects-16-00974-t009:** Main differences between *H. cingula* and *H. nigrifascia*.

Characters	*H. cingula*	*H.* *nigrifascia*
Posterior mac on Th. II	23	37–45
Mac on Th. III	30	43–53
Mac on Abd. I	9	9–11
Central mac on Abd. II	5	6
Central mac on Abd. III	1	1–2
Central mac on Abd. IV posteriorly	4–5	6 (7)

## Data Availability

The original contributions presented in this study are included in the article. Further inquiries can be directed to the corresponding author.

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
