# Peer review of "A Review of the Genus Homidia (Collembola, Entomobryidae) in China Informed by COI DNA Barcoding, with the Description of Three New Speciesâ€"

_insects, 2025, doi:10.3390/insects16090974_

Round 1
Reviewer 1 Report
Comments and Suggestions for Authors
The title does not reflect the content of the paper. First, the paper reports essentially an integrative taxonomy of some Homidia species from China. Three new species are described, and some specific names may be synonyms, but there is no actual review of the genus in China. Second, the paper does not include phylogenetic consideration (revealing the events of cladogenesis of the genus).
The authors stated they used maximum likelihood for tree reconstruction, but the results showed only a neighbour joining tree.
The tree they have generated is barely a NJ tree of the single COI marker, thus could only reflect the distance-based relationship among the COI genes of the studied species, but not the phylogeny of them.
The node labels on the tree only show genetic distances between nodes, but not supports of the nodes, thus the robustness of the reconstruction can not be determined. The node supports should be provided.
In several places the authors stated ‘phylogeny’ or ‘phylogenetic’, however, those are essentially about genetic distances but not phylogeny. The authors should remove all the indications about phylogeny in the manuscript because they have not actually done it.
If the authors want to formally synonymize H. laha with H. sauteri, and H. linhaiensis with H. tiantaiensis, the statements should be given more clearly.
Fig 2, Fig 8B and Fig 15B show green or yellow tone that does not reflect the true coloration. The authors should adjust the white balance of these pictures.
The structures of Table 1 and Table 5 should be improved.
Comments on the Quality of English LanguageThe English writing does not meet the standards of international publication, thus needs to be thoroughly improved.
Author Response
in the attached word

Reviewer 2 Report
Comments and Suggestions for Authors
- There are many problems with the English. I have made many suggestions to improve the narrative, but the manuscript has to be further reviewed for language. Most changes just complete sentences, but in a couple of instances I was at a loss trying to figure out what the authors were trying to say. My suggested changes to the narrative are included in the pdf attached.
- I suggest the title be changed. This is not really a review of Chinese Homidia in the usual sense of the term.
- The main conceptual problem has to do with the phylogenetic analysis and tree. a) the tree placement in the manuscript is not the best, it should be closer to the section where it is discussed. b) the methods state that tree was constructed using maximum likelihood, but the caption to the figure states it is a NJ tree. Neighbor Joining is a distance method, not a maximum likelihood method, K2-P is a distance correction model, not a likelihood. I have no problem with the NJ tree based on K2-P, but the last sentence in the methods section should be deleted. c) Since the authors include a scale for branch lengths at the base of the tree the numbers on the tree branches are unnecessary, I think they should be deleted d) There is no discussion related to the tree, in fact, it is unclear why the authors include a tree if there is no stated framework for its purpose. The authors discuss the synonymy of 2-3 species, presumably based on the results of the phylogenetic analysis, but the discussion is vague and full of gaps. The tree could be discussed more fully to justify species delimitations.
- I do not understand the meaning of the gray band in Table 5. Please explain clearly.

The English needs much improvement. The manuscript is difficult to read and in a couple of instances the translation from Chinese into English doesn't work at all. The species descriptions themselves are generally fine because they are standardized descriptive language, but the rest of the manuscript really needs to be reviewed by someone with better English writing skills.
Author Response
In the attached Word

Round 2
Reviewer 1 Report
Comments and Suggestions for Authors
The title is better phrased as: A Review of the Genus Homidia (Collembola, Entomobryidae) in China informed by COI DNA Barcoding, with the Description of Three New Species
Line 17: some species are discussed should be rephrased as: the taxonomic status of some species are discussed
Line 33-35: The references in which the characters are proposed should be given.
Line 41: Sixteen of Homidia have been reported should be: sixteen species of Homidia have been reported or sixteen Homidia species have been…
Line 410: slightly should be slight
Line 433-434: What does ‘the numbers of individual and species were the most common’ mean? Should it be ‘the numbers of individual and species were the greatest’?
The species H. mediofascia was not included in either molecular or morphological analyses of this study, why did the authors discuss it at the end of the paper? I do not think the statement is supported by their results.
Author Response
Dear reviewer,
Thank you very much for your review.
I revise the MS according to your good suggestions one by one.
Best wishes.
Yours
Yitong MA